# Single-Cell DNA Methylation Analysis of Chicken Lampbrush Chromosomes

**DOI:** 10.3390/ijms232012601

**Published:** 2022-10-20

**Authors:** Artem Nurislamov, Timofey Lagunov, Maria Gridina, Alla Krasikova, Veniamin Fishman

**Affiliations:** 1Department of Natural Sciences, Novosibirsk State University, Pirogova St. 1, 630090 Novosibirsk, Russia; 2Institute of Cytology and Genetics SB RAS, Lavrentiev Avenue 10, 630090 Novosibirsk, Russia; 3Department of Physics, Novosibirsk State University, Pirogova St. 1, 630090 Novosibirsk, Russia; 4Laboratory of The Cell Nucleus Structure and Dynamics, Cytology and Histology Department, Saint Petersburg State University, Universitetskaya Embankment 7/9, 199034 Saint Petersburg, Russia; 5Artificial Intelligence Research Institute (AIRI), Kutuzovskiy Avenue 32, 121108 Moscow, Russia

**Keywords:** DNA methylation, chicken oocyte, chicken genome, lampbrush chromosomes

## Abstract

DNA methylation is an essential epigenetic regulation mechanism implicated in transcription and replication control, developmental reprogramming, retroelements silencing and other genomic processes. During mammalian development, a specific DNA methylation pattern should be established in germ cells to allow embryonic development. Less is known about germ cell DNA methylation in other species. To close this gap, we performed a single-cell methylome analysis of chicken diplotene oocytes. We comprehensively characterized methylation patterns in these cells, obtained methylation-based chicken genome segmentation and identified oocyte-specific methylated gene promoters. Our data show that despite the formation of specific transcriptionally hyperactive genome architecture in chicken diplotene oocytes, methylation patterns in these cells closely resemble genomic distribution observed in somatic tissues.

## 1. Introduction

DNA methylation in vertebrates is implicated in endogenous and exogenous gene silencing, heterochromatin formation, establishment, and maintenance of chromatin architecture, DNA repair, and other genomic processes. In contrast to other epigenetic modifications, DNA methylation can be inherited through cell divisions, allowing the transfer of epigenetic information across the mitotic cycle in somatic cells. To reset this epigenetic memory, there are specific mechanisms governing epigenetic reprogramming and the establishment of germ-line-specific methylation patterns in germ cells and early embryos.

During mammalian development, sperm progenitors undergo two rounds of epigenetic reprogramming, and at the end of gametogenesis, the mature sperm genome exhibits ~80–90% CpG methylation. The genomic distribution of DNA methylation in sperm is roughly similar to that of somatic cells, although with a slightly higher DNA methylation content [1]. In oocytes, methylation gradually increases during development, starting from practically unmethylated non-growing oocytes present in the primordial follicle before folliculogenesis, and reaching the level of ~40–50% in mature gametes. The establishment of methylation patterns in oocytes depends on transcription, thus methylated regions are mainly restricted to the transcribed gene bodies, leaving intergenic regions and non-transcribing genes hypomethylated. Overall, mammalian gametes show specific patterns and levels of DNA methylation, which is required to maintain the developmental program of these species [2].

There is substantially less information about DNA methylation in gametes of non-mammalian vertebrate species. Here, we focus on epigenetic profiling of germ cells in domestic chicken, Gallus gallus. Chicken is one of the most characterized non-mammalian vertebrate species, and chicken models play a pivotal role in animal research as an alternative and outbreed experimental species. In addition, domestic chicken is the globally most important source of commercially produced meat. Chicken models were extensively studied using modern genomic approaches, providing rich data about genome architecture [3], transcription dynamics [4,5], chromatin accessibility [6], and methylation patterns [7] in somatic cells. Moreover, DNA methylation in chicken sperm cells was recently profiled, showing that sperm DNA is hypomethylated, which correlates with the absence of the DNMT3L cofactor in the chicken genome [7]. No genomic data are available to study DNA methylation in growing chicken oocytes. During the diplotene stage of prophase I of meiosis, the chicken oocyte genome acquires specific organization, characterized by elevated transcription and formation of lampbrush chromosome structures. Although cytological evidence suggests that DNA methylation in growing chicken oocytes correlates with chromatin organization and transcription, there are no molecular data available [8,9].

Here, we profile DNA methylation in two stages of chicken oocyte diplotene. We show that oocyte methylation at both stages shows patterns previously observed in somatic cells, including hypomethylation of CpG islands and hypermethylation of transposons and other repetitive DNA elements. In contrast to sperm, average levels of DNA methylation in oocytes are on par with somatic cells. We developed a new HMM-based segmentation tool that identifies hypomethylated regions in the chicken genome and showed that there is a limited number of oocyte-specific hypomethylated gene promoters. Overall, our data suggest that DNA methylation does not significantly contribute to the developmental reprogramming of the oocyte and lampbrush chromosome formation in birds.

## 2. Results

### 2.1. Chicken Oocytes Isolation and Bisulfite Sequencing

To profile oocyte DNA methylation in adult chickens and study its dynamics, we chose SWF (small white follicles) and LWF (large white follicles) stages, which can be easily distinguished using microscopy analyses. Both SWF and LWF represent diplotene stages, roughly similar to 10–15 days post-partum stages of mammalian oocyte development [2]. We defined SWF and LWF stages based on the follicle size (1–2 mm follicle for the SWF stage, 4–7 mm follicle for the LWF stage). To ensure that follicles with different sizes contain oocytes at different developmental stages, we isolated oocyte nuclei and accessed their size and chromosome morphology (Figure 1). During the SWF stage, a typical avian diplotene nucleus can be observed—with lampbrush bivalents evenly distributed across the nucleus (Figure 1A), which corresponds to previously published data [10]. Nuclei at later stages slightly increase in size, but the main difference between SWF and LWF stages is chromosome morphology. At the LWF stage, chromosomes tend to gather together, and their morphology is altered to a far more condensed state (Figure 1C) [11]. Thus, we confirmed that the stratification of oocytes by follicle size allows for unambiguous discrimination between lampbrush and post-lampbrush stages.

In order to avoid contamination by follicular cells surrounding the oocyte, we isolated the oocyte nuclei as described previously [12]. We performed bisulfite sequencing of 16 oocyte nuclei from SWF and LWF stages (7 SWF and 9 LWF oocytes collected from 3 animals) generating from 3 to 7 million read pairs for each oocyte which results in coverage of 2–4 million CpG dinucleotides per oocyte (Appendix A). Merged data for all oocytes allowed us to access methylation in 69.73% of CpG dinucleotides. Our optimized mapping setup for bismark allowed us to reach average mapping efficiency of 69.7% which is on par with existing single-cell bisulfite sequencing data on mammalian oocytes [13].

Given the fact that during the diplotene stage there are four chromatids of each chromosome and each converted DNA strand is sequenced independently, we can expect no more than eight unique reads covering each CpG dinucleotide. Thus, cases, where dinucleotides are covered more than eight times, can be considered artifacts or contamination from somatic cells. Our data show that there is a very small amount (~0.1–0.2%) of artifacts across all libraries (Figure 2A). Thus, our approach for bisulfite sequencing of individual chicken oocyte nuclei performs on par with existing single-cell approaches by showing high mapping efficiency and reproducibility.

### 2.2. Methylation Profile of Chicken Oocytes

To compare levels and patterns of DNA methylation in oocytes with other chicken cell types we uniformly processed publicly available data for fibroblasts [14], spleen, jejunum, ileum [7], muscles [15], and sperm cells [16]. We found that the average level of CpG methylation in oocytes (53.2%) was comparable to those observed in somatic cells (52.45–62.8%). We also confirmed that the sperm genome is hypomethylated (40.5%), as was previously reported [7]. 

We next compared methylation levels on micro- and macro-chromosomes, and methylation patterns of different genomic features. In general, there are no major differences in DNA methylation patterns between chicken lampbrush chromosomes and somatic cells (Figure 2B). Promoters and CpG islands showed bimodal distribution: whereas the majority of these loci were hypomethylated, a small fraction displayed high methylation levels. Exons also showed a bimodal distribution of methylation, where the majority of the loci were highly methylated and few percentages of exons showed no methylation. Similarly, 3′-untranslated regions were highly methylated, whereas promoter-proximal 5′-untranslated regions were almost completely demethylated. Intergenic sequences, repetitive elements and introns showed increased methylation levels (Figure 2C). 

We did not detect any substantial difference between micro- and macrochromosomes, except slightly higher average methylation levels on macrochromosomes, which could be explained by a higher portion of introns, repetitive elements and intergenic features. Sperm cells show decreased methylation levels for all genomic features analyzed, confirming again the previously described hypomethylation of the sperm genome (Figure 2D). 

We next measured the methylation of all 5-kb genomic bins and compared the methylation of individual loci between chicken cell types. This analysis showed a moderate correlation between methylation profiles in oocytes and somatic cells (Spearman’s R~0.6–0.7, Figure 3A). We noticed a difference between hypomethylated (<40%) and hypermethylated (>60%) loci: the latter always show more dissimilarities between cell types (except muscle cells that differ in methylation pattern from all somatic cells; Appendix A). A pairwise comparison of cell types showed that methylation level within hypermethylated loci is less consistent between cell types than in hypomethylated regions (Appendix A). Nevertheless, methylation of the CpG islands and gene promoters, which often fall within hypomethylated regions, was very similar. Moreover, this methylation pattern does not reflect known features of oocyte-specific transcription. For example, the histone genes cluster was demethylated in oocytes similarly to somatic cell types (Appendix A), despite the absence of histone locus bodies and inactivation of the histone gene cluster [10]. HoxD gene cluster contains an extremely long (~80-kb, chr7:16,335,000–16,407,000) demethylated region, which was observed in all analyzed cell types (Appendix A). 

Since lampbrush chromosomes display unique chromatin organization compared to somatic cell interphase nuclei, we examined whether the methylation of CTCF binding sites differs between diplotene oocytes and fibroblasts. CTCF is a major vertebrate insulator that plays an important role in genome folding by mediating the formation of chromatin loops [17,18]. Reportedly, domain borders in lampbrush chromosomes do not correspond with somatic TAD boundaries [19]. However, we did not find any significant differences in levels of DNA methylation between oocytes and fibroblasts in CTCF sites suggesting that there are different mechanisms involved in chromatin domain insulation in lampbrush chromosomes (Appendix A).

It is known that some repetitive sequences, such as CNM, PO41, and Z-microsatellite are transcriptionally active during the lampbrush stage [20], whereas others, such as chromosome-specific centromeric repeats Cen1, Cen2, Cen3, Cen4, Cen7, Cen8, and Cen11, are transcriptionally silent [21]. Due to their repetitive nature, methylation analysis of these sequences cannot be performed using standard genomic alignment. We manually designed sequences representing methylated and demethylated states of these repetitive elements after bisulfite conversion. Next, we aligned reads using these references and counted a portion of alignments corresponding to the methylated and demethylated states. The results of this analysis (Appendix A) showed that all repeats except PO41 display relatively high methylation levels (50–60%). However, the distinctive results of PO41 methylation (10–20% in most of the cell types) should be interpreted with caution, because this repetitive element was covered by a minimal amount of reads (~10 reads in each cell type). We did not observe any correlation between the transcription status of the repeat and its methylation: actively transcribed CNM and Z-microsatellite repeats show relatively the same methylation level as non-transcribed Cen-repeats. Interestingly, methylation levels of all repeat sequences in oocytes and sperm cells were lower than in somatic cells.

Finally, we compared methylation of SWF and LWF stages of oocyte development. We found no difference in methylation levels between the stages (*p*-value = 0.11); moreover, the difference between individual animals was substantially higher than the variation between different oocyte stages (Appendix A). We did not observe any difference in the methylation patterns for the analyzed genomic features, such as promoters, enhancers, and CpG islands, specific for SWF or LWF. Thus, we concluded that, in contrast to mammals, there are no changes in methylation during SWF to LWF oocyte transition. Overall, our data showed that chicken oocytes at both SWF and LWF stages display a methylation pattern very similar to those previously described in chicken and mammalian somatic cells.

### 2.3. Methylation-Informed Segmentation of the Chicken Genome Identifies Oocyte-Specific Hypomethylated Regions

In contrast to the limited difference in methylation patterns between cell types, we observed well-pronounced differences in methylation levels between different loci (Figure 3B). As was mentioned above, low methylation was often attributed to promoters and CpG islands; however, the overlap between demethylated regions and these genomic features was not complete (the quantification of the overlap size is provided below).

To identify differentially methylated genomic regions, we developed a segmentation method based on the Hidden Markov model (HMM). We modified standard multinomial HMM to account for the unequal distance between neighboring CpG sites and unequal coverage of each CpG site (see a more detailed description of modifications in Methods). Iterating over model parameters showed that the observed methylation levels should be explained by two or three different methylation regimes, therefore we used three methylation regimes for all subsequent studies. These regimes, according to the average methylation levels of the corresponding loci, were assigned to hypo-, intermediate- and hypermethylated regimes (Figure 3C). 

Genomic segmentation obtained using HMM corresponds well to the visual observations (Figure 3B). All three methylation regimes can be well-discriminated based on the average methylation level of the locus; however, the difference between hypomethylated and other methylation regimes was the most pronounced. The vast majority of hypomethylated loci show methylation level that does not exceed 20% (Appendix A). We identified more than 40,000 hypomethylated loci in oocytes, with a mean size of ~650 b.p. (the exception was chromosome Z, where the mean hypomethylated loci length exceeds 1 kb). There was slight, although significant enrichment of gene promoters across hypomethylated loci (656 vs. 455 ± 21 expected at random), and more than two-times enrichment of CpG islands (634 vs. 286 ± 19 expected at random), suggesting that hypomethylated loci represent regulatory genomic elements. However, many hypomethylated loci do not match any known promoter or CpG island. 

Consistent with our previous results, loci identified as hypomethylated in oocytes often show low average methylation levels in other cell types. Using a genome segmentation obtained on oocyte data and a threshold of 20% CpG methylation as a discriminative feature of the hypomethylated loci, we compared the distribution of hypomethylated loci in oocytes and other cell types. We defined oocyte-specific loci as regions that are either hypomethylated in oocytes but not in other cell types, or hypomethylated in all cell types but not oocytes. Note that in this analysis we excluded sperm cells from the comparison because the methylation of these cells strongly differs from other cell types (Figure 2B). Using this definition, we found ~197 loci specific for oocytes (Appendix A). 

Only 656 of the identified loci overlapped promoters, and after visual inspection, we filtered out many of them based on the signal quality (Appendix A). This results in 194 genes that show oocyte-specific patterns of methylation in promoters. Interestingly, several genes implicated in early development showed distinct methylation profiles in oocytes. For example, the promoter area of the NKX2-6 gene (homeobox-containing early development transcription regulator) contained an oocyte-specific hypomethylated region (Figure 3D and Appendix A), whereas for KLF4 (pluripotency-associated transcription factor) we observed oocyte-specific gain of promoter methylation. Moreover, we identified an oocyte-specific decrease in promoter methylation of the TET2 (Tet Methylcytosine Dioxygenase 2) gene. Increased expression of this demethylation enzyme may be responsible for oocyte-specific hypomethylation. However, gene ontology analysis on the complete set of oocyte-specific hypomethylated promoters did not show any enrichment of developmental terms. This, again, suggests that methylation patterns observed in oocytes are largely consistent between cell types and there is only a limited number of loci showing oocyte-specific methylation.

## 3. Discussion

Here, we present a comprehensive characterization of chicken lampbrush-stage oocyte methylome. Computational analysis and concordance between our results and previous cytological analyses suggest a high quality of the obtained data. For example, a high level of methylation of transposons and other repetitive DNA correlates with previous cytological data where the distribution of 5-methylcytosine along the axes of chicken lampbrush chromosomes was characterized by immunostaining [9]. Enrichment of 5mC was demonstrated in compact chromomeres associated with the clusters of certain tandem repeats, leading to the general repression of these genomic elements during oogenesis [8].

To our surprise, we did not find a substantial genome-wide difference between oocyte and somatic cells methylome, despite the specific genome architecture of diplotene oocytes. All differences that we observed were limited to a small number of loci, and the number of differentially methylated loci was on par with a variability of somatic cell types. Similarly, transcriptional silencing that occurs during SWF to LWF transition does not change the methylation pattern that appeared to be very similar at SWF and LWF stages. Thus, we concluded that epigenetic factors other than DNA methylation are responsible for lampbrush chromosome formation. Among these factors—hyperacetylation of histone H4 was observed not only on the transcription loops but also at places where they are anchored into chromomeres [9,22]. Increased chromatin accessibility followed by initiation of hypertranscription may lead to the appearance of visible extended lateral loops. In addition, we note that these observations are completely opposite to what can be seen for mammals, where during the oocyte growth phase, DNA methylation increases and imprinted loci acquire a specific methylation profile. 

It was suggested that CpG methylation along lampbrush chromosomes can carry certain epigenetic information and regulate gene expression after fertilization [8]. However, it was not known whether the DNA methylation profile is preserved after bivalent condensation. Although limited in number, we identified several gene promoters that showed oocyte-specific methylation profiles. Presumably, demethylation of transcription start sites can act as labeling for activation of early development genes during zygotic activation. Further epigenetic and transcriptomic studies of SWF, LWF and later oocyte stages can be performed to test this hypothesis.

Finally, we highlight the new HMM-based approach that we developed to segment the genome onto differentially methylated states. Although HMM is a well-established genome segmentation technique [23], here we implemented some specific modifications that are important for methylation data analysis. This allowed us to produce methylation-based segmentation of the chicken genome. Although the obtained segmentation was based on the oocytes’ data, due to the relatively small variability of methylation levels between cell types, we assume that this segmentation can be used as a reference in future research focused on chicken genomics.

## 4. Materials and Methods

### 4.1. Nuclei Isolation

Nuclei were extracted from oocytes of domestic chicken with 1–2 mm (SWF stage) and 4–7 mm (LWF stage) diameter by using a modified protocol [11]. Oocytes were isolated from the chicken ovary and transferred to a drop of “5:1+” buffer (83.0 мМ KCl, 17.0 мМ NaCl, 6.5 мМ Na_2_HPO_4_, 3.5 мМ KH_2_PO_4_, 1.0 мМ MgCl_2_, 1.0 мМ DTT), tungsten needle was used to make holes in oocyte envelope, then exposed nuclei were carefully transferred to a clean drop with a micropipette to wash away the remaining yolk. All animal experiments were approved by the bioethics committee of the Institute of Cytology and Genetics SB RAS (Protocol №66, 9 October 2020). International guidelines were followed during experimental procedures (“Guide for the Care and Use of Laboratory Animals” [24]).

### 4.2. Staining of Oocyte Nuclei and Confocal Microscopy

Isolated nuclei were transferred to 2% PFA diluted in “5:1” buffer (83.0 мМ KCl, 17.0 мМ NaCl, 6.5 мМ Na_2_HPO_4_, 3.5 мМ KH_2_PO_4_, 1.0 мМ MgCl_2_) and incubated for 15 min for fixation. PFA was quenched by incubating nuclei for 15 min in a 0.125M solution of glycine diluted in a “5:1” buffer. Fixed nuclei were transferred to individual wells of the 96-well black microplate (Eppendorf, Hamburg, Germany) and kept in 45 µL of PBS. Prior to microscopy, SYTOX Green stain (Invitrogen, Waltham, MA, USA) was added to a final concentration of 0.07 µM.

Confocal laser scanning microscopy was performed by using LSM 780 (Zeiss, Jena, Germany) inverted confocal system supplied with a 488 mm argon laser and Plan-Apochromat 20x/0.8 M27 objective. A Series of optical slices was captured with 1024 × 1024 format followed by 3D reconstruction and image maximum projection views in ZEN Black software (Zeiss, Jena, Germany). Additional processing of digital images was performed by using ImageJ 1.53 (National Institutes of Health, Bethesda, MD, USA) and Photoshop CC 2021 (Adobe, San Jose, CA, USA) software.

### 4.3. Library Preparation and Sequencing 

For the preparation of single-cell bisulfite sequencing libraries, we used a modified version of a default protocol for the Pico Methyl-Seq Library Prep Kit (Zymo Research, Irvine, CA, USA). Each isolated nuclei in a volume of 3 µL were transferred from a drop of “5:1” buffer to individual tubes with 16 µL of nuclease-free water and 0.8 units of Proteinase K (New England Biolabs, Ipswich, MA, USA) and incubated for 2 h at 65 °C. After incubation, 130 µL of Lightning Conversion Reagent was added. Further steps were performed according to standard protocol with adjustments: incubation time with L-Desulphonation Buffer was reduced to 17 min, PrepAmp primers final concentration was reduced to 2 µM and the amount of amplification cycles with LibraryAmp primers was set to 10.

Libraries were quantified using a Qubit fluorometer (Invitrogen, Waltham, MA, USA). Sequencing was performed on the Illumina platform in 150 nt paired-end modes in the ICG SB RAS.

### 4.4. Data Analysis

#### 4.4.1. Raw Data Preparation

For data analysis, the Bismark-0.23.0 software was used with the Bowtie2-2.4.4 aligner. R1 and R2 sequences were aligned separately in non-directional mode with linear minimum-score function (bismark --bowtie2 --score_min L,0,-0.7 --non_directional). Next, two bam files with aligned reads were combined and deduplicated (deduplicate_bismark --bam --multiple R1.bam R2.bam). After that, an additional script was used to remove all alignments with a mapping quality of less than 10. The filtered bam file was used for extracting methylation (bismark_methylation_extractor --comprehensive --bedGraph --CX filtered.bam). Finally, the coverage2cytosine (coverage2cytosine --ff --genome_folder {genome} filtered.cov.gz) function was used to generate the CpG_report.txt and CpG.cov files, which were used in further analysis.

#### 4.4.2. HMM

We modified the python HMM implementation proposed by [25]. We modeled the HMM state emission as a Binomial distribution function Bin(*nt*,*βk*), where *nt*—*t*’s CpG site read-depth and *βk*—average methylation level of HMM state *k*. In addition, the Binomial distribution function has been raised to the power of (*nt* + 2) during HMM training so that cytosines with greater read depth contribute more to HMM learning. These modifications were based on [26]. Additionally, we modified transition probability to model the decrease in a correlation of methylation levels with increasing genomic distance [27]):(1)$$P\left(S_{t}|S_{t-1}=k,d_{t-1}\;\;\right)=\left\{\begin{array}{c} \frac{1}{4}-\frac{1}{4}\exp\left(-\frac{d_{t-1}}{d_{0}}\right),\;S_{t}\neq S_{t-1}\\ \frac{1}{4}+\frac{3}{4}\exp\left(-\frac{d_{t-1}}{d_{0}}\right),\;S_{t}=S_{t-1} \end{array}\right.$$
where *S_t_*—state of CpG site *t*, *S_t−_*_1_—state of CpG site *t* − 1, *k*—state number, *d_t−_*_1_—distance from CpG site *t −* 1 and *t*, *d*_0_—distance parameter.

In contrast with standard multinomial HMM that works with directive data series (such as time-dependent events), the methylation state of *t*’s CpG site should be connected with both *t* − 1 and *t* + 1 CpG site states. To take this into account, we add the reverse-directed data in the training set and decoding set. After the whole data were decoded, we intersect the forward and reverse decodings and named all CpG sites with a difference in forward and reverse states as “un” (unknown) and excluded them from further analysis.

The developed MethylationHMM implementation can be obtained at https://github.com/TimaLagunov/methylation-decoder (accessed on 17 September 2022).

All data sets were prepared by the set_maker.py script. For parameter optimization, the train and test subsamples of the oocyte data were used. To define train and test subsets, we selected 54 non-overlapping loci, 300 kb length each, with randomly chosen start positions so that all three types of chicken chromosomes (macro-chromosomes, inter-chromosomes and micro-chromosomes) had the same representation in train and test sets. The AIC and BIC metrics from HMM implementation [25,28] were used to find the optimal parameters for MethylationHMM (n_states = 3, distance_param = 1250). These optimal parameters were used to make further decoding of oocyte methylation states.

### 4.5. Data Availability

Sequencing data for chicken oocytes are available via NCBI BioProject PRJNA834620. Bisulfite data for somatic cells and sperm cells were downloaded from SRA (SRR14664594, SRR11278879, SRR11278872, SRR11278852, SRR5015166, SRX514415).

## Figures and Tables

**Figure 1 ijms-23-12601-f001:**
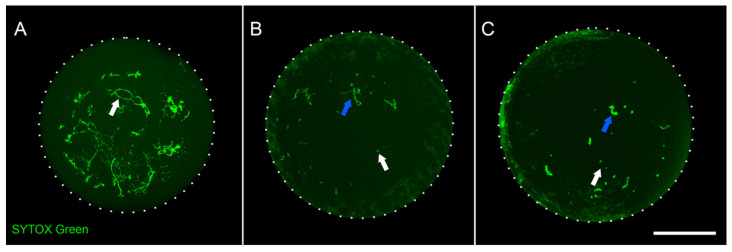
Follicle size allows discrimination of developmental stages of chicken oocytes. (**A**)—nucleus from 2 mm stage follicle (SWF). White arrow indicates lampbrush chromosomes. (**B**)—nucleus from 3 mm stage follicle. White arrow indicates microchromosome, blue arrow indicates macrochromosome. (**C**)—nucleus from 4 mm stage follicle (LWF). White arrow indicates condensed microchromosome, blue arrow indicates condensed macrochromosome. Staining with Sytox Green followed by confocal laser scanning microscopy; maximum intensity projection images are shown. Scale bar—100 μm.

**Figure 2 ijms-23-12601-f002:**
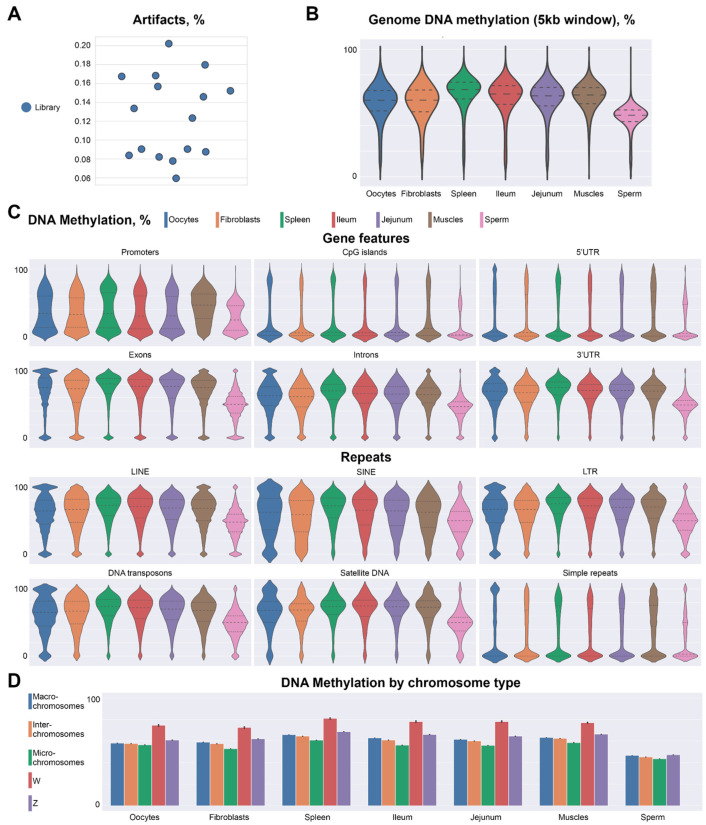
Chicken diplotene oocytes display similarities with somatic cells in DNA methylation patterns across chromosome types and genomic features. (**A**)—artifacts across all single-cell libraries. (**B**)—comparison of genome methylation levels between chicken oocytes and other cell types. (**C**)—comparison of methylation levels across genomic features and repeated elements between oocytes and different cell types. (**D**)—DNA methylation for different chromosome types.

**Figure 3 ijms-23-12601-f003:**
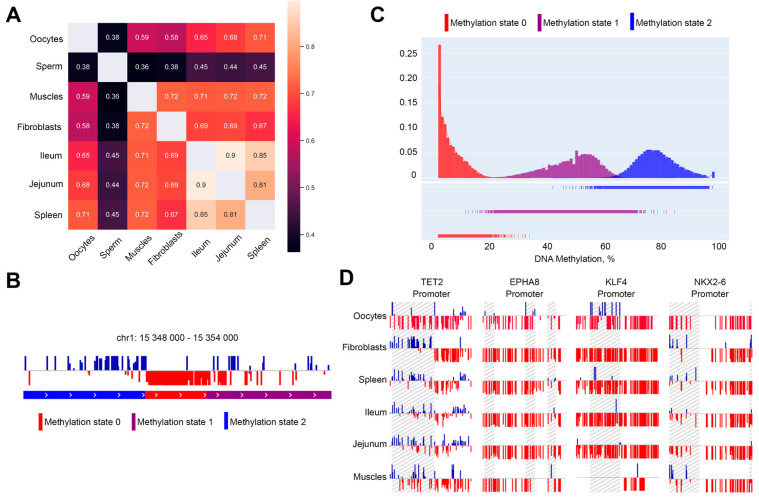
Identification of differentially methylated regions. (**A**)—correlation between methylation features between different cell types. (**B**)—example of methylation states calculated by Hidden Markov model. States 0, 1, 2 represent hypo-, intermediate- and hypermethylated regions. (**C**)—distribution of different methylation states. (**D**)—examples of differentially methylated regions within promoters identified using HMM. Differentially methylated regions are highlighted.

## Data Availability

The data presented in this study are available in Methods section.

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
