# Peer review of "Single-Cell DNA Methylation Analysis of Chicken Lampbrush Chromosomes"

_ijms, 2022, doi:10.3390/ijms232012601_

Round 1
Reviewer 1 Report
It’s an interesting work, but some issues are confusing. The followings are questions and suggestions:
1. Methylation of SWF and LWF stages of oocyte development were compared, and the difference between individual animals was substantially higher than variation between different oocyte stages (Supplementary Figure 5). However, it’s confusing that SWF were from chicken1&3 while LWF were from chicken 2&3. There’s no difference between SWF and LWF in chicken 3, but some information was lost in chicken 1&2. It’s not rigorous enough. Please explain it.
2. Are SWF and LWF mentioned in line 94 from one chicken? Please give more details as there are differences between individuals.
3. More analysis angles and verification can be added to make the content more plentiful.
4. According to the average methylation levels of the corresponding loci, were assigned to hypo-, intermediate- and hyper- methylated regimes (line 196). Is there a clear range? How to define hypo-, intermediate- and hyper- ?
Author Response
Response to Reviewer 1 Comments
Reviewer 1:
It’s an interesting work, but some issues are confusing. The followings are questions and suggestions:
Point 1. Methylation of SWF and LWF stages of oocyte development were compared, and the difference between individual animals was substantially higher than variation between different oocyte stages (Supplementary Figure 5). However, it’s confusing that SWF were from chicken1&3 while LWF were from chicken 2&3. There’s no difference between SWF and LWF in chicken 3, but some information was lost in chicken 1&2. It’s not rigorous enough. Please explain it.
Response 1. While we agree that isolation of both LWF and SWF oocytes from each animal would be the best option, we note that this is challenging due to the peculiar properties of these cells. Nuclei isolation is a very delicate process and processing each cell requires large time. After isolation of ovaries, oocytes degrade rapidly, and there is a small time window while nuclei can be collected. This usually limits the number of nuclei to 5 or 6 from each animal. We started this experiment with chicken 1 (SWF oocytes) and chicken 2 (LWF oocytes), and after data analysis we suspected that there might be some difference in methylation levels between SWF and LWF oocytes. In order to check that the difference between oocytes is tied to their stage and to further expand the number of samples, we took a challenge to simultaneously isolate nuclei from both SWF and LWF stages of oocytes using chicken 3. Bisulfite libraries from chicken 3 resulted in similar methylation levels at both stages. Thus, we concluded that the difference between individual animals was higher than variation between different oocyte stages.
Point 2. Are SWF and LWF mentioned in line 94 from one chicken? Please give more details as there are differences between individuals.
Response 2. All 16 oocytes from our experiments were collected from 3 animals. We updated the manuscript and added this information. Additional information regarding the number of oocytes collected from individual animals is presented in Supplementary Figure 6.
Updated text in the manuscript: “We performed bisulfite sequencing of 16 oocyte nuclei from SWF and LWF stages (7 SWF and 9 LWF oocytes collected from 3 animals) generating from 3 to 7 million read pairs for each oocyte which results in coverage of 2-4 million CpG dinucleotides per oocyte (Table S1)”
Point 3. More analysis angles and verification can be added to make the content more plentiful.
Response 3. Following your suggestion regarding performing additional analysis, we added comparison of DNA methylation in CTCF binding sites between oocytes and fibroblasts.
Updated text in the manuscript: ”Since lampbrush chromosomes display unique chromatin organization compared to somatic cell interphase nuclei, we examined whether the methylation of CTCF binding sites differ between diplotene oocytes and fibroblasts. CTCF is a major vertebrate insulator that plays an important role in genome folding by mediating formation of chromatin loops [17,18]. Reportedly, domain borders in lampbrush chromosomes do not correspond with somatic TAD boundaries [19]. However, we did not find any significant differences in DNA methylation between oocytes and fibroblasts in CTCF sites suggesting that there are different mechanisms involved in chromatin domain insulation in lampbrush chromosomes (Figure S4)”
Point 4. According to the average methylation levels of the corresponding loci, were assigned to hypo-, intermediate- and hyper- methylated regimes (line 196). Is there a clear range? How to define hypo-, intermediate- and hyper- ?
Response 4. The Hidden Markov Model (HMM) was used to identify the methylation regimes. We set the number of regimes (3) as hyperparameter for this model, but the most likely regime for each CpG site was found by the HMM. After the HMM marked all CpG sites (as regime 0, 1 or 2), we analyzed the methylation levels observed for each regime (Figure 3C). According to the obtained distributions, methylation regimes was named as hypo-, intermediate- and hyper-. It can be seen from Figure 3C that the methylation levels in hypo-methylated regime has a very small intersection area with the intermediate-methylated regime (in contrast with the intermediate-hyper intersection area). So, we can set a threshold between hypo-methylated and intermediate-methylated regimes at ~20% of methylation (and less strict threshold between intermediate- and hyper- regimes at ~65% of methylation). Note that the methylation regimes don’t have the clear range of methylation levels because the methylation regime of i-th CpG site depends not only from the methylation levels observed in this site, but also from the methylation level of its neighbors (i+1 and i-1).
On behalf of all authors,
Veniamin Fishman
Reviewer 2 Report
The manuscript is devoted to the study of one of the components of epigenetic regulation, DNA methylation, of lamp brush-type chromosomes in chicken oocytes at different stages of development. Interesting data have been obtained, which can be published in the International Journal of Molecular Science. It has been shown that, unlike mammalian oocytes and avian spermatozoa, the methylation profile of chicken oocytes is generally similar to that in somatic cells. This indicates that the change of stages of oocyte development is under the control of mechanisms other than DNA methylation. The reviewer has only minor comments:
Figures 2 and 3 do not have a common name for the entire figure, which is usually present, but only the names of individual parts of the figure. It is advisable to add a common name for the entire panel of figures to improve perception.
Author Response
Response to Reviewer 2 Comments
Reviewer 2:
The manuscript is devoted to the study of one of the components of epigenetic regulation, DNA methylation, of lamp brush-type chromosomes in chicken oocytes at different stages of development. Interesting data have been obtained, which can be published in the International Journal of Molecular Science. It has been shown that, unlike mammalian oocytes and avian spermatozoa, the methylation profile of chicken oocytes is generally similar to that in somatic cells. This indicates that the change of stages of oocyte development is under the control of mechanisms other than DNA methylation. The reviewer has only minor comments:
Point 1. Figures 2 and 3 do not have a common name for the entire figure, which is usually present, but only the names of individual parts of the figure. It is advisable to add a common name for the entire panel of figures to improve perception.
Response 1. As per your advice, we added common names for Figures 2 and 3.
Updated text in manuscript: “Figure 2. Chicken diplotene oocytes display similarities with somatic cells in DNA methylation pattern across chromosome types and genomic features”
“Figure 3. Identification of differentially methylated regions”
On behalf of all authors,
Veniamin Fishman
Reviewer 3 Report
Abstract: No comments
Introduction: No comments
Results: In line 136-137, high correlation mentioned with R~ 0.6-0.7, is this really a high correlation or a moderate one?
Also in line 137-138, "We noticed a difference between hypomethylated (<40%) and hypermethylated (>60%)", is this difference significant?
Discussion: No comments
Materials and Methods: No comments
Reviewer 4 Report
In this manuscript, Nurislamov and colleagues present single-cell DNA methylation data from chicken lampbrush chromosomes. The take-home message of this study is that DNA methylation in these chromosomes, which are obtained from oocytes undergoing meiotic prophase, is not much different from somatic tissues. The study is well conducted and the quality of the presentation acceptable. However, I feel that biological significance is still missing. Please find my comments below.
1. Biological significance:
The authors found various promoter gene regions differentially methylated in oocytes. This is interesting and could provide biological significance to this study. Hence, RT-PCR for the best candidate genes or ideally RNA-seq is required.
2. List of genes affected:
Please include the list of ~200 hypomethylated genes in oocytes.
3. Fig. 3D:
Is the data shown coming from a single clone or the average of multiples ones? Please clarify.
In addition, showing several clones displaying same result in Supp. Figures would be useful.
4. Number annotations in text and tables:
Please use point as decimal separator instead of comma.
Author Response
Response to Reviewer 4 Comments
Reviewer 4:
In this manuscript, Nurislamov and colleagues present single-cell DNA methylation data from chicken lampbrush chromosomes. The take-home message of this study is that DNA methylation in these chromosomes, which are obtained from oocytes undergoing meiotic prophase, is not much different from somatic tissues. The study is well conducted and the quality of the presentation acceptable. However, I feel that biological significance is still missing. Please find my comments below.
Point 1. Biological significance:
The authors found various promoter gene regions differentially methylated in oocytes. This is interesting and could provide biological significance to this study. Hence, RT-PCR for the best candidate genes or ideally RNA-seq is required.
Response 1. Chicken oocyte transcriptome consists of RNA in nuclei, cytoplasm, and transcripts in oocyte’s cytoplasm imported from follicular cells (mostly rRNA, doi: 10.1007/s00441-016-2444-4). While we can isolate nuclei from oocytes, we are still facing two issues. First, nuclei are embedded in a dense yolk layer (especially in LWF and later stages), which makes it difficult to completely eliminate cytoplasmic RNA (it is possible to remove most of it, but we expect that background noise may still appear). Complete separation of nuclear and cytoplasmic RNA at lampbrush stage is possible in amphibians due to less yolk and bigger nuclear size, the latter allows to open the nuclear envelope and extract its contents (doi: 10.1101/gad.202184.112).
Second, it’s reported that avian oocytes can accumulate transcripts from multiple stages (doi: 10.1017/S0967199499000398). This may cause confusing results, since not much is known about expression and DNA methylation patterns during early stages of avian oogenesis. Although nascent RNA-seq could possibly expand our knowledge, it will require a time consuming optimization of existing protocols for chicken oocytes, which is not a standard research object for such applications.
Point 2. List of genes affected:
Please include the list of ~200 hypomethylated genes in oocytes.
Response 2. According to your suggestion, we included a list of differentially methylated regions in Supplementary table 4 and added reference to this table in the text.
Updated text in manuscript: “Only 656 of the identified loci overlapped promoters, and after visual inspection we filtered out many of them based on the signal quality (Table S4). This results in 194 genes that show oocyte-specific pattern of methylation in promoters.”
Point 3. Fig. 3D:
Is the data shown coming from a single clone or the average of multiples ones? Please clarify.
In addition, showing several clones displaying same result in Supp. Figures would be useful.
Response 3. Data shown in Figure 3D represents average DNA methylation levels of multiple oocytes (as stated in this figure - “Oocytes”). As per your advice, we included additional Supplementary figure 7 with single-cell examples.
Updated text in manuscript: “For example, promoter area of NKX2-6 gene (homeobox containing early development transcription regulator) contained oocyte-specific hypomethylated region (Figure 3D, Figure S7), whereas for KLF4 (pluripotency-associated transcription factor) we observed oocyte-specific gain of promoter methylation.”
Point 4. Number annotations in text and tables:
Please use point as decimal separator instead of comma.
Response 4. As per your advice, we updated all decimal separators to points both in text and supplementary material.
On behalf of all authors,
Veniamin Fishman
Round 2
Reviewer 4 Report
I am satisfied with the authors's response.